# A Scientometric Study to a Critical Review on Promising Anticancer and Neuroprotective Compounds: Citrus Flavonoids

**DOI:** 10.3390/antiox12030669

**Published:** 2023-03-08

**Authors:** Mingyang Qiu, Wenlong Wei, Jianqing Zhang, Hanze Wang, Yuxin Bai, De-an Guo

**Affiliations:** 1College of Pharmacy, Changchun University of Chinese Medicine, Changchun 130117, China; 2Shanghai Research Center for Modernization of Traditional Chinese Medicine, National Engineering Research Center of TCM Standardization Technology, Shanghai Institute of Materia Medica, Chinese Academy of Sciences, Shanghai 201203, China

**Keywords:** citrus flavonoids, scientometric analysis, anti-cancer, neuroprotection, mechanism of action

## Abstract

Flavonoids derived from citrus plants are favored by phytomedicinal researchers due to their wide range of biological activities, and relevant studies have been sustained for 67 years (since the first paper published in 1955). In terms of a scientometric and critical review, the scientometrics of related papers, chemical structures, and pharmacological action of citrus flavonoids were comprehensively summarized. The modern pharmacological effects of citrus flavonoids are primarily focused on their anticancer activities (such as breast cancer, gastric cancer, lung cancer, and liver cancer), neuroprotective effects (such as anti-Alzheimer’s disease, Parkinson’s disease), and metabolic diseases. Furthermore, the therapeutic mechanism of cancers (including inducing apoptosis, inhibiting cell proliferation, and inhibiting cancer metastasis), neuroprotective effects (including antioxidant and anti-inflammatory), and metabolic diseases (such as non-alcoholic fatty liver disease, type 2 diabetes mellitus) were summarized and discussed. We anticipate that this review could provide an essential reference for anti-cancer and neuroprotective research of citrus flavonoids and provide researchers with a comprehensive understanding of citrus flavonoids.

## 1. Introduction

Citrus genome studies have shown that citrus plants originated in the Himalayas about 6 to 8 million years ago, and the majority of species are descendants of wild broad-hued oranges (*C. reticulata*), grapefruit (*C. maxima*), and citron (*C. medica*) [1]. As time went on, people found citrus fruits were not only beautiful and edible, but also fairly tasty. In a bid to improve the taste of citrus fruit, people have developed several citrus cultivation varieties, such as *C. hystrix*, *C. japonica*, C. *mitis*, *C. aurantifolia*, *C. paradisi*, *C. junos*, *C. limetta*, *C. maximas*, *C. aurantium*, *C. limon*, and *C. sinensis*. In recent years, citrus has become one of the most productive fruits in the world for economic cultivation [2]. Since ancient times in China, Egypt, and India, citrus fruit was not only used as a tasty fruit but also as a medicine. The dried peel of *Citrus reticulata Blanco* was used as an ingredient in tea and in Chinese patent medicines, which was beneficial to promote health by regulating “qi”. Modern botanists are also interested in the medicinal value of citrus. Botanical medicine researchers have found that citrus fruits are rich in a variety of beneficial components, such as fibers [3], phenolic acids [4], and flavonoids [5]. Notably, citrus flavonoids, recognized as a class of substances with important nutritional value, are comprehensively investigated. Citrus flavonoids are usually classified according to their chemical structures, such as flavanone aglycones, flavone aglycones, flavanonols, flavanone-O-glucoside, polymethoxyflavonoids, other flavonoids, flavone-O-glucoside, and flavone-C-glucoside [6]. Studies have shown that citrus flavonoids possess a variety of pharmacological properties including antioxidant and and anti-inflammatory [7], among others. In view of the complex chemical composition and diverse pharmacological activities of citrus flavonoids, we believe that it is significant to summarize Citrus flavonoids by means of scientometric analysis and a critical review.

The scientometric methods were first applied to obtain a holistic and comprehensive view based on the published studies on citrus flavonoids, which is an application of mathematical and statistical methods to perform retrospective reviews, calculate correlations in publication data, elucidate current research progress, and predict research directions [8]. Scientometric methods have played an important role in bone disease research [9], hotspots of exercise for intervening diabetes [10], COVID-19 research [11], and exosome studies [12]. The numerous published articles were summarized through the scientometric methods for providing rich reference information for researchers in need based on keywords, highlights, and important research-related information. 

The research protocol of this article was shown in Figure 1. We summarized the development of scientific research on citrus flavonoids, the chemical structure, and distribution of citrus flavonoids in plants, and summarized the main pharmacological actions including antioxidation, inhibition of cancer progression, and neuroprotection. The mechanisms of these pharmacological actions were also discussed. The purpose of this article was to sort out the research history of citrus flavonoids, the plant sources of citrus flavonoid compounds, and the main pharmacological mechanisms of action of citrus flavonoids in order to provide researchers with a comprehensive understanding of citrus flavonoids.

## 2. Materials and Methods

### 2.1. Literature Search and Data Download

Citrus flavonoid was searched as a topic in the Web of Science core collection database as of 19 July 2022. Full records and cited references considered as raw data were downloaded from the database, and the file format was plain text.

### 2.2. Scientometric Analysis and Visualization

CiteSpace was initially used for bibliometric analysis; the factors include country, institute, keyword, category, reference, and cited journal, with parameter settings of time slicing (1991–2022), node type, and selection criteria (top 50 levels of the most cited or occurring items). In addition, VOSviewer was applied to optimize and provide an aesthetic map. Impact factor (IF) and Hirsh Index (H-index) were fully considered for a comprehensive and scientometric analysis.

## 3. Results and Discussion

### 3.1. Scientometric Analysis

#### 3.1.1. General Analysis

As a result, 3202 publications about citrus flavonoids were downloaded from the Web of Science core collection database as of 19 July 2022. Most of the articles were published in 2021 (Figure 2a) and the most popular category was food science technology (Figure 2b). Studies on citrus flavonoids have been continuously increasing in number.

#### 3.1.2. Journal Analysis

Twenty-five journals published the majority of articles related to citrus flavonoids. As shown in Table 1, the *Journal of Agricultural and Food Chemistry* had the highest publication number, followed by *Food Chemistry*, *Molecules*, *Food and Function*, and the *Journal of Functional Foods*. Interestingly, as shown in Table 2, five journals possessed fewer publications but higher citations (cited number > 600), which were the *Journal of Biological Chemistry*, *Life Sciences*, the *Journal of Nutrition*, *PLoS ONE*, and *Phytochemistry*.

#### 3.1.3. Country/Region and Institution Analysis

China published the highest number of articles, followed by USA, Italy, Japan, South Korea, Spain, India, and Brazil (Figure 2c), among which the first two were the most cited countries (Figure 2d). Figure 2e presents the top 10 countries with the strongest citation bursts, indicating that the studies on citrus flavonoids in Saudi Arabia have increased in the past two years. 

The United States Department of Agriculture (USDA) was the most influential institution with the highest number of publications, followed by Egyptian Knowledge Bank Ekb, University of Messina, Consejo Superior De Investigaciones Cientificas Csic, Jeju National University, Zhejiang University, Rutgers State University, New Brunswick, and Southwest University China (Figure 2f). There were positive correlations between the countries, research institutions, and the areas of citrus cultivation. The most cited institution was the University of Messina (Figure 2g).

#### 3.1.4. Literature and Cited Reference Analysis

Figure 3a presents the document citation co-occurrence review. The top 15 most cited articles on citrus flavonoids are shown in Table 3. The most cited article reported that vitamin C and phenols were the main components of antioxidant capacity in citrus juice [13]. The second most cited article reported the extraction of polyphenols, especially flavanones from orange (*Citrus sinensis* L.) peel by using ethanol as a food-grade solvent [14]. Figure 3b presents a co-citation reference review. The top 15 cited references on citrus flavonoids are shown in Table 4. 

#### 3.1.5. Keywords Analysis 

The keyword co-occurrence network is shown in Figure 3c. The weight of the circle size denotes the frequency of occurrence of keywords. The keyword population distribution is shown in Figure 3d. The top 15 keywords with the strongest citation bursts are shown in Figure 3e.

### 3.2. Chemical Structures and Sources of Citrus Flavonoids

The structures of citrus flavonoids are summarized in Figure 4. These compounds could be classified by structural type as flavanone aglycones, flavone aglycones, flavanonols, flavanone-O-glucoside, polymethoxyflavonoids, flavone-O-glucoside, flavone-C-glucoside, and other flavonoids. Flavanone aglycones constitute the parent nucleus of flavanone glycosides. They comprise a class of compounds derived from the parent nucleus of 2-phenyl dihydrochromone. 2-Phenyl dihydrochromone is a flavanone which is also known as dihydroflavone. The representative compounds of flavanone aglycones are hesperidin, naringin, dihydroquercetin, and isosakurin, respectively. Flavone aglycone refers to the parent nucleus of flavonoid aglycone, and its main components are acacia, 8-hydroxyapigenin, luteolin, kakaol, five-hydroxyl flavone, apigenin, and geraniol. Flavanonol compounds are the double bonds of flavonoids at the C2 and C3 positions hydrogenated to form flavanonols, and those with hydroxyl groups at the C3 position are generally called flavanonols, and there are three compounds in citrus flavonoids of this type. Flavanone-O-glucoside refers to the O-glycoside formed by the connection between the sugar substituents and flavanone carbon skeleton in the form of hydrogen oxidation. The typical compounds are hesperidin and naringin. Polymethoxyflavonoids are a kind of constituent exclusive to citrus plants. They contain multiple methoxyls which possess low polarities, flat structures, and strong biological activity. The representative components of polymethoxyflavonoids are tangeretin and nobiletin. Other flavonoids are *Myrica rubra* flavone and catechol. Flavone glycosides are the most abundant substances among Citrus flavonoids. According to the different linking modes between aglycones and sugar molecules, they could be divided into oxyside and carboside. The main components of the flavone glycoside type are vitzene-2, diosmin, and rutin.

The literature on Citrus plant components was summarized, and the distribution of various flavonoids in citrus plants is shown in Figure 5. *C. hystrix* only contains yukovanol; *C. japonica* only contains flavone-O-glucoside and flavone-C-glucoside; *C. medica* and *C. mitis* contain flavanone-O-glucoside and flavone glycosides; *C. aurantifolia* contains flavanone-O-glucoside, flavone glycosides, and polymethoxyflavonoids; *C. sinensis*, *C. limon*, *C. maximas*, and *C. paradisi* include all structure types; *C. junos* contains all structural types except flavanone aglycones; *C. reticulata* includes all structural types and contains the most flavone glycosides among this type of compound; *C. limetta* contains only flavanone aglycones and flavone glycosides. The abovementioned results indicate that citrus plants tend to enrich flavonoids in fruits rather than roots. Naringin, narirutin, hesperidin, and rutin are the most widely distributed compounds found in citrus plants. 

### 3.3. Citrus Flavonoids and Cancers

According to the search results, citrus flavonoids have an obvious inhibitory effect against various cancers [18], the most studied of which is breast cancer; other cancers include rectal cancer, gastric cancer, liver cancer, lung cancer, prostate cancer, uterine cancer, ovarian cancer, epidermal cancer. Citrus flavonoids play a role in cancer therapy by inhibiting cancer cell proliferation [40], migration, angiogenesis, and inducing apoptosis [41]. We summarized the molecular mechanism of citrus flavonoids in cancer therapy and provided a reference for cancer therapy research.

It should be noted here that CYP3A4, as the most important oxidative enzyme, plays a metabolic role for most drugs [42], and the role of CYP3A4 on drug metabolism in cancer treatment has attracted more attention [43]. While grapefruit inhibits the expression of CYP3A4 [44], some studies have shown that *Fructus aurantia* and tangeretin induce CYP3A4 [45,46]. It is clear that some citrus flavonoids have a regulatory effect on CYP3A4. It is suggested that we should be cautious when consuming products derived from citrus to avoid reducing the efficacy of the drug or enhancing the adverse effects.

#### 3.3.1. Breast Cancer

Compounds and molecular mechanisms of citrus flavonoids against breast cancer are shown in Figure 6.

Nobiletin, a natural flavonoid isolated from citrus peel, has anti-angiogenic effects [47]. Nobiletin was shown to inhibit MCF7 breast cancer cells by inducing its metabolism by up-regulating cytochrome P450 family 1 subfamily A member 1 (CYP1A1) and cytochrome P450 family 1 subfamily B member 1 (CYP1B1) [48]. Furthermore, nobiletin was shown to induce apoptotic cell death by reducing B-cell leukemia/lymphoma 2 xL (Bcl-xL) expression without affecting Bcl-2-associated x protein (Bax) levels and inhibit the activities of protein kinase B (AKT) and downstream mammalian target of rapamycin (mTOR) [49]. These targets are located in the apoptosis pathway, suggesting that the treatment of breast cancer by tangerine is mainly through inducing the apoptosis of cancer cells. 

Naringenin was shown to inhibit the growth of metastases after surgery by modulating host immunity [50]. Naringin can inhibit cancer cell reproduction by inhibiting vascular endothelial factor release [51] and regulating the β-catenin pathway [52]. Hesperetin can induce apoptosis in breast cancer cells by triggering the accumulation of reactive oxygen species (ROS), activating the apoptosis signal-regulating kinase 1 (ASK1)/c-jun n-terminal kinase (JNK) pathway, and activating targets of caspase-9 and caspase-3. Hesperetin could increase the Bax: B-cell lymphoma-2 (Bcl-2) ratio in the intracellular environment [53]. The results indicate that hesperetin can inhibit cancer cells by inducing apoptosis. Polymethoxyflavonoids was shown to induce apoptosis in breast cancer cells [54] by activating a Ca (2^+^)-dependent pro-apoptotic protease [55]. Retusin and Ayanin are potent inhibitors of breast cancer resistance protein (BCRP), showing only slightly lower potency than Ko143 [56]. 

According to the abovementioned results, it is known that the therapeutic mechanism of citrus flavonoids affecting breast cancer mainly depends on inducing apoptosis. In addition, citrus flavonoids inhibit cell proliferation pathways and slow breast cancer cell reproduction. Citrus flavonoids can also inhibit the metastasis of cancer cells. Overall, citrus flavonoids treat breast cancer in a variety of ways. In conclusion, the therapeutic effect of citrus flavonoids on breast cancer is clear. However, more comprehensive and in-depth studies are needed to make citrus flavonoids a suitable drug for the treatment of breast cancer.

#### 3.3.2. Colorectal Cancer

Nobiletin showed a strong inhibitory effect on the growth of colon cancer cells [57] by inhibiting matrix metallopeptidase 7(MMP-7) (Figure 7) gene expression [58]. Nobiletin inhibited cancer invasion and metastasis by increasing tissue the tissue inhibition of metalloproteinase-1 (TIMP-1) production [59]. In addition, it was found that nobiletin could down-regulate leptin levels [60]. High levels of leptin in mice are thought to be a key factor in promoting colorectal cancer. Tangerine was metabolized in the intestine to 3′-desmethylnorchol, 4′-desmethylnorchol and 3′,4′-didemethylnorchol. These metabolites were considered to be key compounds for the treatment of intestinal cancer [61].

Other citrus flavonoids such as tangeretin can induce cell cycle G1 arrest [62], and hesperidin can promote cancer cell apoptosis through Caspase-3 (CASP3) activation [63]. 

Naringenin and hesperetin play a critical role in inhibiting the formation of abnormal crypt foci [64] and reducing the activity of bacterial enzymes in colon cancer [65]. 

Nobiletin is an important component for the treatment of colon cancer. The mechanism of citrus flavonoids on colon cancer can induce apoptosis of cancer cells, inhibit the growth of cancer cells, and regulate intestinal enzymes. It is believed that the most important mechanism is still the induction of apoptosis. Additional research is needed to clarify the mechanism of citrus flavonoids in the treatment of colon cancer.

#### 3.3.3. Gastric Cancer

Nobiletin could inhibit proliferation and induce apoptosis of gastric cancer cells [66]. Nobiletin could also slow the progression of cancer by extending the cell growth cycle [67]. The preliminary effect of naringenin in treating gastric cancer has been demonstrated [68] by inhibiting cell proliferation, migration, and invasion [69], and by causing ASK1-induced apoptosis mediated by ROS (Figure 7) [70].

#### 3.3.4. Lung Cancer 

Citrus juice rich in beta-cryptoxanthin and hesperidin could inhibit lung tumor growth in mice [71]. Tangeretin suppresses interleukin-1 (IL-1) beta-induced cyclooxygenase (COX)-2 expression through inhibition of p38, mitogen-activated protein kinase (MAPK), JNK, and AKT activation in human lung carcinoma cells (Figure 7) [72].

Nobiletin inhibited the epithelial–mesenchymal transition (EMT) of human non-small-cell lung cancer (NSCLC) cells by antagonizing the Transforming Growth Factor-β1 (TGF-β1)/SMAD Family Member 3 (Smad 3) signaling pathway, which could play a crucial role in inhibiting lung cancer metastasis [73].

5-Demethyltangeretin inhibited human non-small-cell lung cancer cell growth by inducing G2/M cell cycle arrest and apoptosis [74].

Flavanones and 2′-OH flavanones could inhibit the growth of A549 and Lewis lung cancer cells in vivo [75].

Hesperidin produced in vitro inhibitory effects NSCLC cells by modulating immune response-related pathways that affect apoptosis [76]. These results provide scientific support for the use of flavonoids extracted and isolated from citrus plants for the treatment of human lung cancer. 

#### 3.3.5. Liver Cancer

Naringenin induced cell cycle arrest and inhibited the growth of human hepatocellular carcinoma cells [77].

Hesperidin induced apoptosis of human hepatocellular carcinoma (HepG2) cells through mitochondrial and death receptor pathways [78].

#### 3.3.6. Prostate Cancer

Naringenin induced apoptosis of prostate cancer cells by regulating Phosphatidylinositol 3-kinase (PI3K)/AKT and MAPK signaling pathways (Figure 7) [79]. 

Naringenin could also promote deoxyribonucleic acid (DNA) repair and prevent carcinogenesis caused by oxidative damage [80].

#### 3.3.7. Cervical Cancer

Hesperetin exhibited potential anticancer activity in vitro against human cervical cancer cell lines by reducing cell viability and inducing apoptosis [81]. 

Naringin also induced growth inhibition and apoptosis in human cervical cancer HeLa cell lines by activating the nuclear factor kappa-B (NF-κB)/Cyclooxygenase-2 (COX-2) -caspase-1 pathway [82].

#### 3.3.8. Ovarian Cancer

Nobiletin inhibited ovarian cancer cells by secreting key angiogenic mediators such as AKT, HIF-1α, NF-κB, and vascular endothelial growth factor (VEGF) (Figure 7) [83]. 

Tangeretin sensitized cisplatin-resistant human ovarian cancer cells by downregulating the PI3K/AKT signaling pathway (Figure 7) [84].

#### 3.3.9. Epidermal Carcinoma

Studies have shown that citrus flavonoids have anti-proliferative effects in inhibiting human squamous cell carcinoma in vitro [85], and the relevant studies proved that naringenin exerts anti-proliferative effects by inducing ROS generation and cell cycle arrest [86].

### 3.4. Neuroprotective Effects of Citrus Flavonoids

Studies have shown that fruits rich in flavonoids could protect the nervous system [87]. Citrus flavonoids inhibited Alzheimer’s disease by reducing Presenilin 1 (PS1) phosphorylation-dependent amyloid production [88], and hesperidin, hesperetin, and neohesperidin exhibited neuroprotective effects [89]. Eriodictyol induced nuclear translocation of nuclear factor erythroid-2 related factor 2 (Nrf2), enhanced heme oxygenase 1 (HO-1) and NAD(P)H quinone dehydrogenase 1 (NQO-1) expression, and increased intracellular glutathione levels against oxidative stress-induced cell death [90]. We summarized studies on the neuroprotective effects of citrus flavonoid compounds to elucidate their mechanisms of action, as shown in Figure 8.

Nobiletin could stimulate protein kinase A (PKA)-mediated phosphorylation of glutamate receptor 1 (GluR1) receptors in the hippocampus to upregulate synaptic propagation through postsynaptic AMPA receptors [91]. It could also rescue cholinergic neurodegeneration and improve memory impairment in olfactory bulbectomy (OBX) mice by reducing the acetyl cholinesterase (AChE) staining and choline acetyltransferase (ChAT) expression density in the hippocampus [92]. Furthermore, nobiletin improved memory impairment and amyloid beta disease in a transgenic mouse model of Alzheimer’s disease [93]. Additionally, triple transgenic (3xTg)-AD mice were orally administrated with 30 mg/kg nobiletin for 3 months, and the results showed that nobiletin reversed the impairment of short-term memory and recognition memory by reducing soluble amyloid beta 1–40 (Aβ1-40) and ROS levels in the mouse brains [94]. Oral administration of nobiletin reduced tau phosphorylation in the hippocampus of senescence-accelerated P8 (SAMP8) mice [95]. Nobiletin rescued 1-methyl-4-phenyl-1,2,3,6-tetrahydropyridine (MPTP)-induced Parkinson’s in a mouse model, reducing the dopamine level in the striatum and hippocampal CA1 region to prevent motor and cognitive dysfunction [96]. Nobiletin reversed learning disabilities associated with the n-methyl-d-aspartate receptor by enhancing cAMP/PKA/extracellular-regulated protein kinase (ERK) signaling in hippocampal neurons and PC12D cells [97]. These studies suggest that nobiletin has great potential in the study of neuropathic diseases.

Naringin protected nigrostriatal nigrothymic dopaminergic (DA) projections from 6-hydroxydopamine (6-OHDA)-induced neurotoxicity [98]. Naringin conferred an important capacity for DA neurons to produce the glial cell-derived neurotrophic factor (GDNF) [99]. In addition, naringin could improve cognitive performance and attenuate oxidative damage [100]. 

Naringenin exerted anti-inflammatory effects due to its interaction with the p38 signaling cascade and signal transducer and activator of the transcription 1 (STAT-1) transcription factor [101]. Naringenin can also inhibit the release of idiopathic oxide (NO) and pro-inflammatory cytokines in microglia [102].

Hesperidin protected against cognitive impairment by inhibiting the overexpression of inflammatory markers such as NF-κB, nitric oxide synthase (NOS), and cyclooxygenase-2 (COX-2) [103]. Furthermore, hesperidin significantly restored the deficits in non-cognitive nesting abilities and social interaction by attenuating amyloid beta deposition in the brain [104].

Hesperetin attenuated lipopolysaccharide (LPS)-induced neuroinflammation, apoptosis, and memory impairment by regulating the toll-like receptor 4 (TLR4)/NF-κB signaling pathway [105], also could reduce malondialdehyde (MDA) in the hippocampus and inhibit brain oxidative stress [106].

Tangeretin significantly protected striatum–substantia nigra integrity [107] by inhibiting LPS-induced phosphorylation of ERK, N-terminal kinase (JNK), and p38 [108]. Previous studies showed that tangeretin partially inhibited the growth of leukemic HL-60 cells by inducing apoptosis, and exhibited less cytotoxicity to normal lymphocytes [109].

Citrus flavonoids protect the nervous system by fighting against inflammation and protecting the function of nerve cells. It is promising to develop citrus flavonoids as an adjuvant treatment for neurodegenerative diseases such as Alzheimer’s disease, Parkinson’s disease, etc. It is believed that the protective effects of citrus flavonoids on the nervous system is worthy of further and in-depth research.

### 3.5. Citrus Flavonoid and Metabolic Disease

Oral administration of bergamot extract (150 mg containing 16% neohesperidin, 47% neohesperidin, and 37% naringin) for 6 months reduced moderate hypercholesterolemia, low-density lipoprotein, and blood lipids in patients with atherosclerosis [110]. Neohesperidin activated the AMPK pathway for hypoglycemic and exhibited lipid-lowering effects [111]. It was obvious that citrus flavonoids were beneficial in the treatment of metabolic diseases. We summarized the regulatory mechanisms of other citrus flavonoids on metabolic diseases.

Nobiletin attenuated dyslipidemia by preventing hepatic triglyceride (TG) accumulation, reducing very low-density lipoprotein (VLDL) and TG secretion [112], while increasing hepatic and peripheral insulin sensitivity and glucose tolerance, and significantly attenuating atherosclerosis in the aortic sinus hardening [113]. In addition, nobiletin could enhance the circadian rhythm to combat metabolic diseases by intervening in the circadian rhythm network [114].

Naringenin reduced the progression of atherosclerosis by improving dyslipidemia [115], apolipoprotein B (apo B) overproduction, and hyperinsulinemia in high-fat-fed mice [116]. Additionally, naringenin maintained lipid homeostasis [117] by preventing cholesterol-induced systemic inflammation and metabolic dysregulation [118]. Naringenin and hesperetin could lower microsomal triglyceride transfer protein (MTP), acetyl-CoA acetyltransferase 1 (ACAT1), and acetyl-CoA acetyltransferase 2 (ACAT2) to reduce blood lipids [119]. Naringenin supplementation enhanced insulin sensitivity and helped to restore glucose homeostasis in diabetic rats [120]. Naringenin activated PI3K without inducing tyrosine phosphorylation of insulin receptor-substrate-1 (IRS-1) to produce insulin-like effects in vivo [121]. Furthermore, naringenin improved cholinergic function and alleviated oxidative stress against type 2 diabetes-induced memory dysfunction by inhibiting elevated cholinesterase (ChE) activity [122]. Naringenin exerted an anti-diabetic effect by upregulating AMPK. Besides, naringenin had a metformin-like effect that reduced inflammation and cell proliferation [123].

Naringin, the major grapefruit flavonoid, primarily affected the development of atherosclerosis in diet-induced hypercholesterolemia in mice [124] by modulating hepatic acetyl-CoA acetyltransferase (ACAT), aortic vascular cell adhesion molecule-1 (VCAM-1), and monocyte chemoattractant protein-1 (MCP-1) [125]. Other effects were reduced inflammatory cell infiltration, reduced oxidative stress, decreased plasma lipid concentrations, and improved hepatic mitochondrial function in rats [126]. Naringin improved bone properties in ovariectomized mice and exerted estrogen-like activity in rat osteoblast-like (UMR-106) cells [127].

Dietary hesperidin exerted hypoglycemic and hypolipidemic effects in streptozotocin-induced borderline type 1 diabetic rats [128]. 

Polymethoxyflavonoids are a novel flavonoid with cholesterol and triacylglycerol-lowering potential, and elevated levels of polymethoxyflavonoid metabolites in the liver may directly lead to its hypolipidemic effect in vivo [129]. Hesperidin stimulated nitric oxide production in endothelial cells while improving endothelial function and reducing inflammatory markers in patients with metabolic syndrome [130].

Hesperetin alleviated hyperglycemia and dyslipidemia by improving the antioxidant capacity of streptozotocin (STZ)-induced experimental rats [131]. Hesperetin prevented diabetes-induced testicular damage by inhibiting oxidative stress, inflammation, and upregulation of enzymatic and non-enzymatic antioxidants [132].

After oral administration of bergamot extract (BPF) to rats and patients for 30 days, BPF significantly reduced triglyceride levels and blood glucose. Meanwhile, BPF inhibited 3-hydroxy-3-methylglutaryl coenzyme A (HMG-CoA) reductase activity and enhanced reactive vasodilation [113].

After diosmin was administered orally (100 mg/kg/day) for 45 days, both histological and biochemical parameters demonstrated antidiabetic effects in type 2 diabetic rats [133].

Based on the above-provided summary, the effect of citrus flavonoids on blood lipids was confirmed. We believe that further research on citrus flavonoids in metabolic diseases is valuable.

## 4. Conclusions

In conclusion, digesting more citrus fruits in our daily diet is likely to be beneficial to our health. Additionally, Citrus flavonoids play a certain role in anti-cancer effects, neuroprotection, and metabolic regulation. Among the many citrus flavonoids, nobiletin is a promising anti-angiogenic agent with great potential for cancer prevention and treatment. In terms of neuroprotection, nobiletin could improve learning and memory deficits, suggesting its anti-Alzheimer’s disease and Parkinson’s disease potential. Moreover, hesperidin and naringin could prevent Parkinson’s disease-related dopaminergic neuron (DA) degeneration, oxidative damage, and cognitive impairment. In short, citrus flavonoids possess high nutritional and medicinal value.

The production of citrus fruits is abundant around the world, and there is still the problem of insufficient utilization of citrus fruits in juice. Therefore, it is important to study and develop the residues of the industrial production of citrus juice. The existing underutilization problem can in turn provide supplements for human health. Moreover, systematic and in-depth research on the neuroprotective effect of citrus flavonoids should be carried out to comprehensively, deeply, and accurately evaluate the neuroprotective effect of citrus flavonoids. The nutritional value of citrus plants as food is also worth of attention, and the content of citrus flavonoids may be used as an indicator to evaluate the nutritional value. Finally, the effects of the long-term use of citrus flavonoids on our health are worthy of further investigation.

## Figures and Tables

**Figure 1 antioxidants-12-00669-f001:**
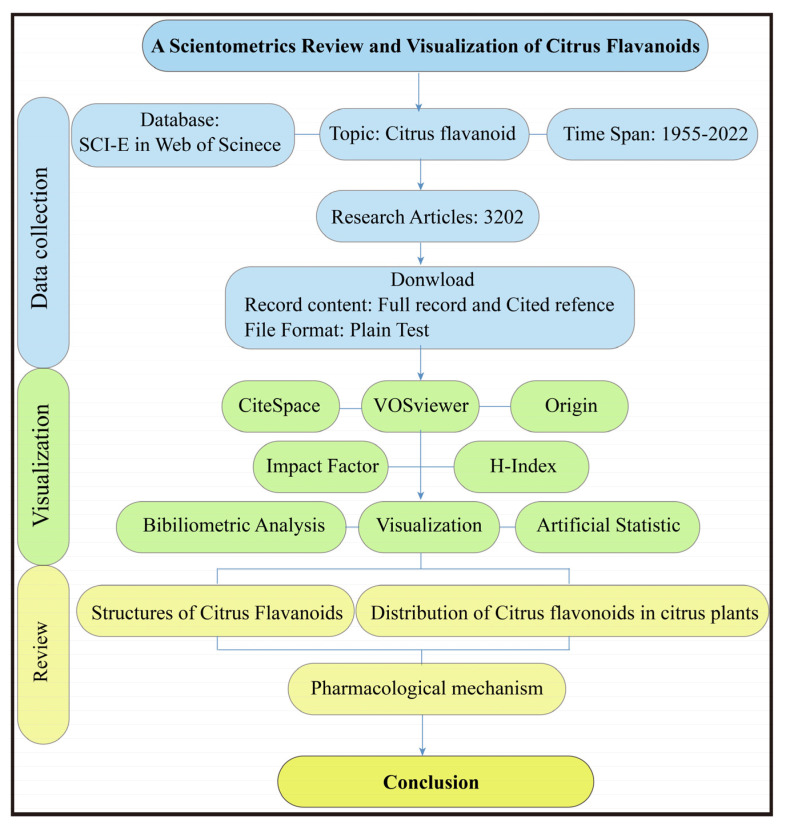
Flow chart of scientometric analysis and review. SCI-E: Science Citation Index Expanded; WOS: Web of Science; IF: Impact Factor; H-Index: Hirsh Index.

**Figure 2 antioxidants-12-00669-f002:**
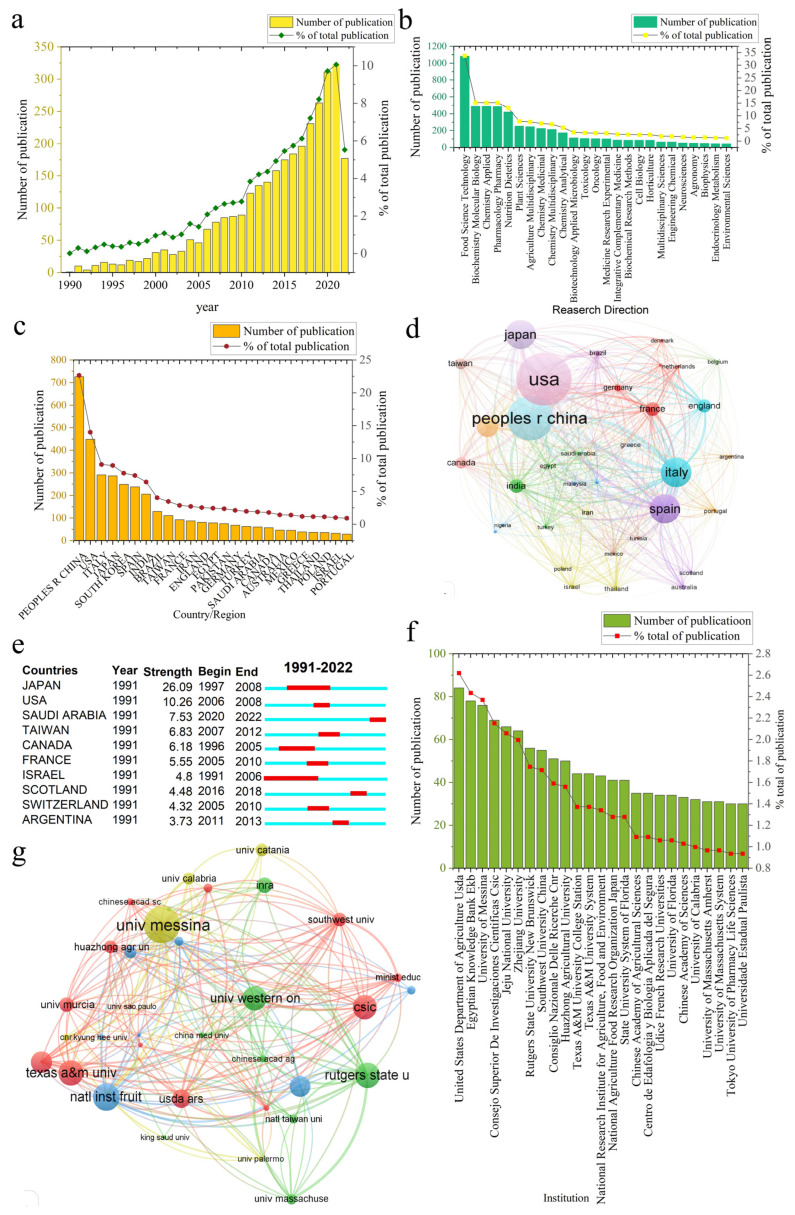
General analysis of the published studies, country/region, and institution analysis. (**a**) Annual publications between 1990 and 2022; the first paper was published in 1955, and the number of published papers was less than five each year from 1955 to 1990, so the figure did not take the number of published papers from 1955 to 1990 into account. (**b**) Category exploration map showing the number of publications on citrus flavonoids in different study directions. (**c**) Number of publications for specific countries/regions. (**d**) Country/region citation co-occurrence view. Node size represents the total number of citations. (**e**) Top 10 countries with the strongest citation bursts. Red bars means that some countries published articles about citrus flavonoids frequently in a certain period. (**f**) Number of publications for specific institutions. (**g**) Institution citation co-occurrence view. Node size represents the total number of citations.

**Figure 3 antioxidants-12-00669-f003:**
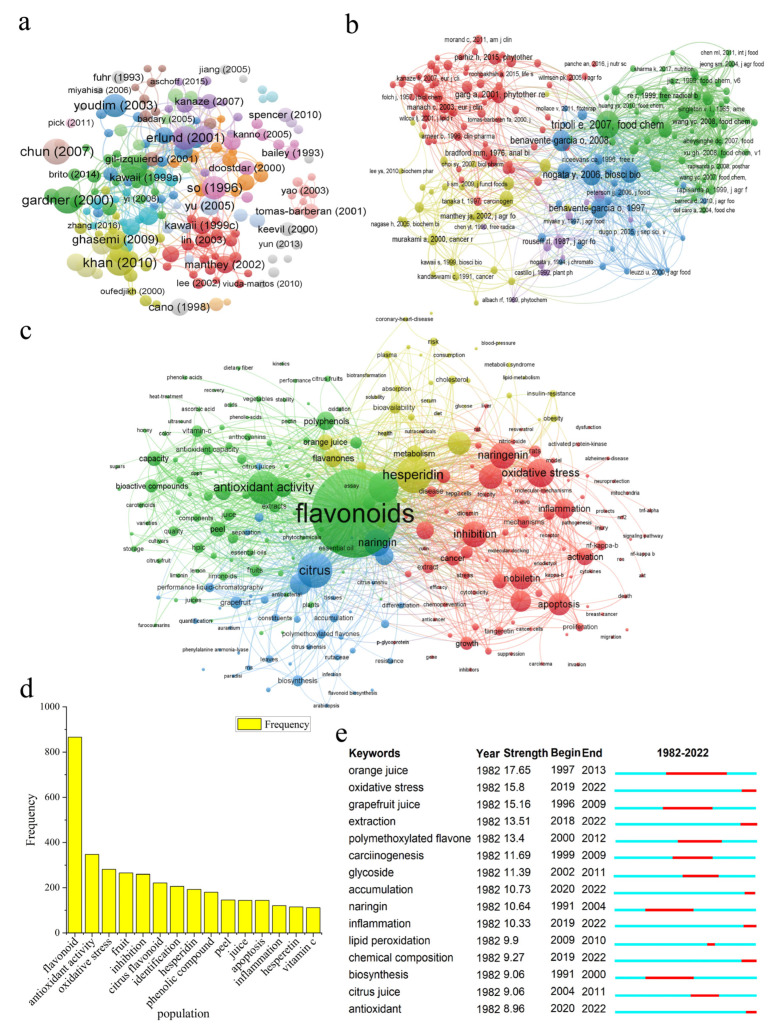
Literature and cited reference co-occurrence review: keyword analysis. (**a**) Document citation co-occurrence review. The minimum number of citations of a document is 100; of the 3202 documents, 179 met the threshold. (**b**) Co-citation reference co-occurrence review. Node size represents the total number of citations. The minimum number of citations of a cited reference is 30; of the 82,378 cited references, 197 met the threshold. For each of the 197 cited references, the total strength of the co-citation links with other cited references was calculated. The cited references with the greatest total link strength were selected. (**c**) Keyword co-occurrence network. Node size and color represent the number of keywords and cluster. (**d**), Keyword population distribution. (**e**) Top 15 keywords with the strongest citation bursts. Red bars denote that some keywords were cited frequently in a certain period.

**Figure 4 antioxidants-12-00669-f004:**
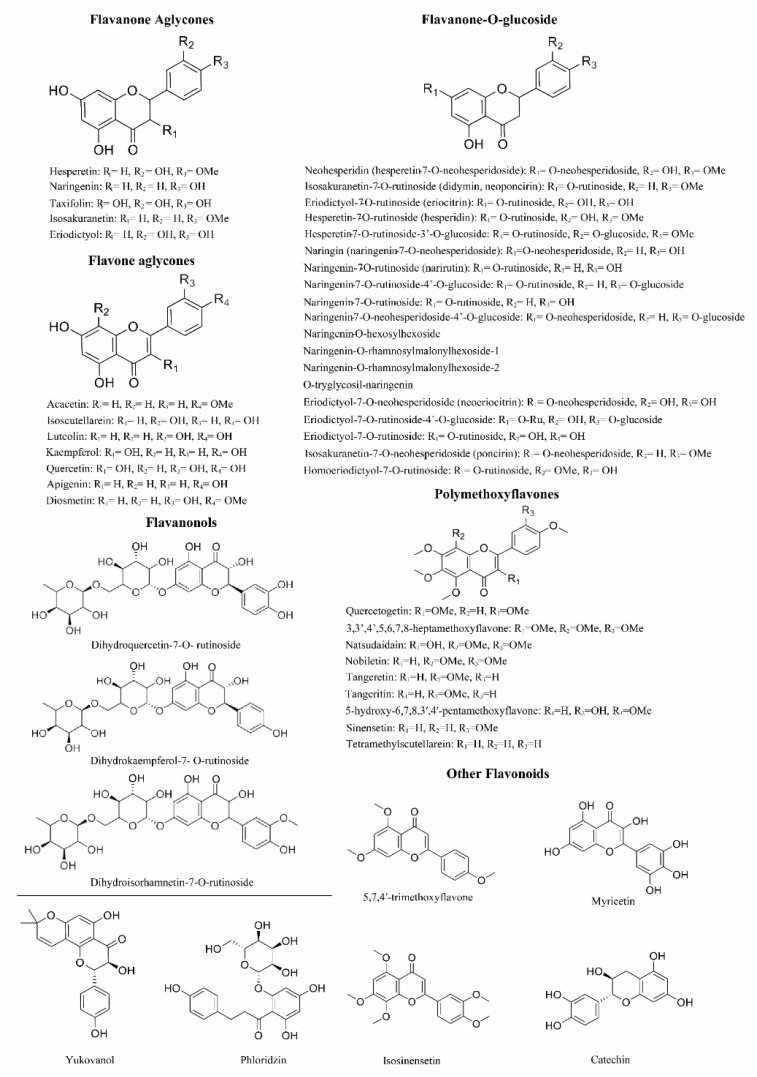
Structures of various citrus flavonoids.

**Figure 5 antioxidants-12-00669-f005:**
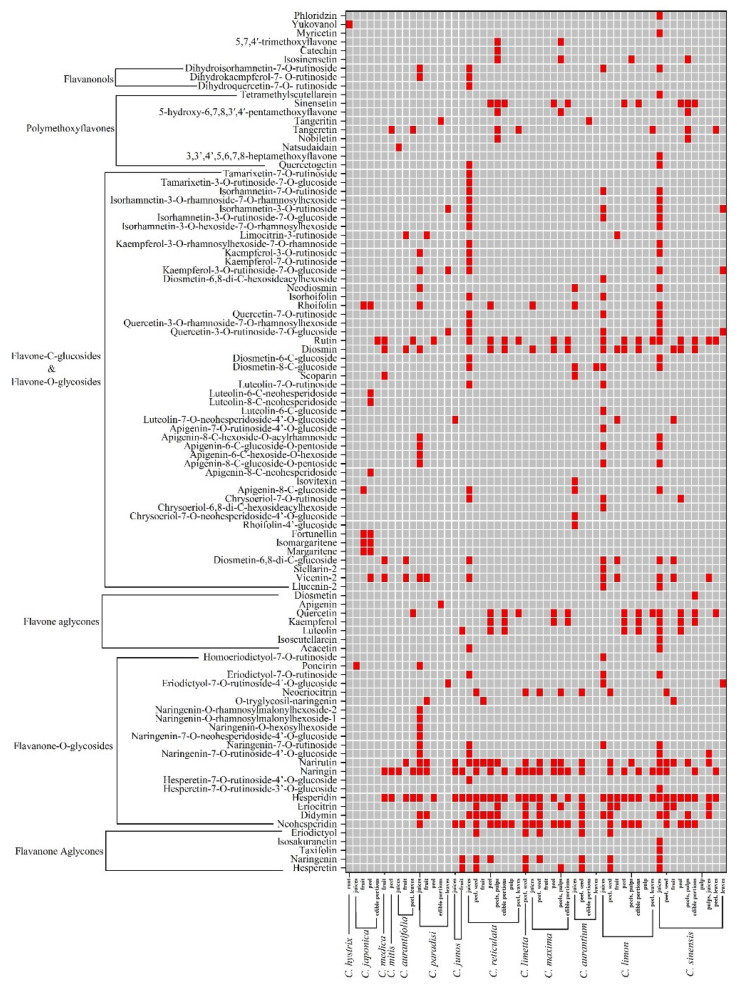
Distribution of citrus flavonoids in 13 citrus plants. Red squares represent compounds present in the corresponding citrus plant parts.

**Figure 6 antioxidants-12-00669-f006:**
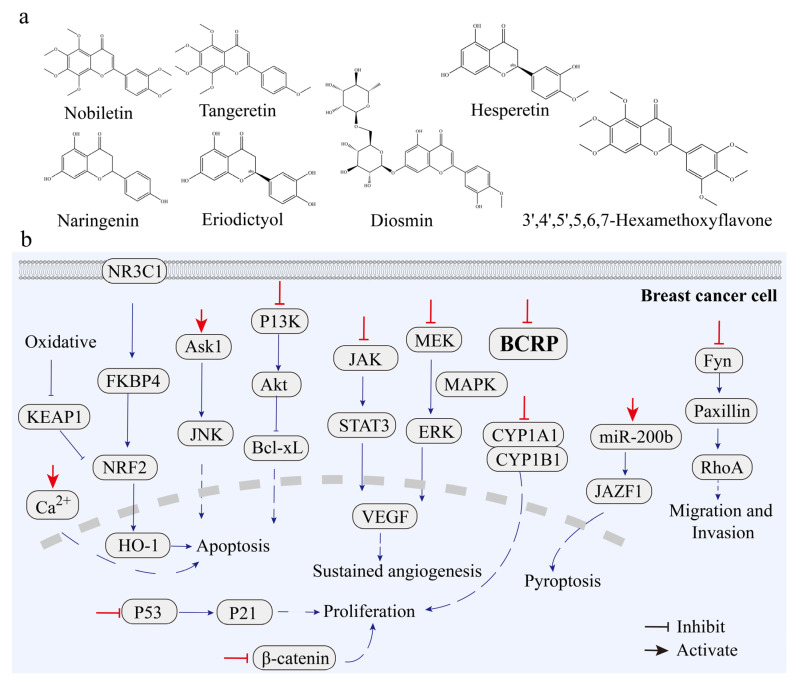
Compounds and molecular mechanisms of citrus flavonoids against breast cancer. (**a**) Citrus flavonoids were used to inhibit breast cancer. (**b**) The possible molecular mechanism of citrus flavonoids inhibiting breast cancer. NR3C1: Nuclear receptor subfamily 3 group C member 1; KEAP1: Kelch-1ike ECH- associated protein I; FKBP4: Monoclonal antibody to FK506-binding protein 4; NRF2: Nuclear factor erythroid 2-related factor 2; HO-1: heme oxygenase 1; P53, P21: tumor suppressor gene; ASK1: Apoptosis signal-regulating kinase 1; JNK: c-Jun N-terminal kinase; PI3K: Phosphatidylinositol 3-kinase; AKT: Protein kinase B; Bcl-xL: B-cell leukemia/lymphoma 2 xL; JAK: Janus kinase; STAT3: Signal transducer and activator of transcription 3; VEGF: Vascular endothelial growth factor; MEK: Methyl ethyl ketone;ERK: Extracellular regulated protein kinases; MAPK: Mitogen-activated protein kinase; BCRP: breast cancer resistance protein; CYP1A1: cytochrome P450 family 1 subfamily A member 1; CYP1B1: cytochrome P450 family 1 subfamily B member 1; miR-200b: microRNA 200b; JAZF1: juxtaposed with another zinc finger gene 1; Fyn: Tyrosine protein kinase FYN; Paxillin: focal adhesions; Rho A: Ras homolog gene family, member ABax: Bcl-2 associated x protein; Bcl-2: B-cell lymphoma-2; P53, P21: Tumor suppressor gene; CDK2: Cyclin dependent kinase 2; CDK4: Cyclin dependent kinase 4; CASP3: Caspase-3; Mcl-1: Myeloid cell leukemia 1; PARP: Poly ADP-ribose polymerase; CASP8: Caspase-8; PI3K: Phosphatidylinositol 3-kinase; Akt: Protein kinase B; Survivin: Inhibitor of apoptosis; Ask1: Apoptosis signal-regulating kinase 1; JNK: c-Jun N-terminal kinase; COX-2: Cyclooxygenase-2; MEK: Methyl ethyl ketone; ERK: Extracellular regulated protein kinases; MAPK: mitogen-activated protein kinase; HIF-1α: Hypoxia-inducible factor 1-alpha; VEGF: Vascular endothelial growth factor; AP-1: Activator protein 1; proMMP7: Proenzyme matrix metallopeptidase 7; MMP7: Matrix metallopeptidase 7; TGF-β1: Transforming growth factor beta-1; Smad3: SMAD family member 3; EMT: Epithelial–mesenchymal transition.

**Figure 7 antioxidants-12-00669-f007:**
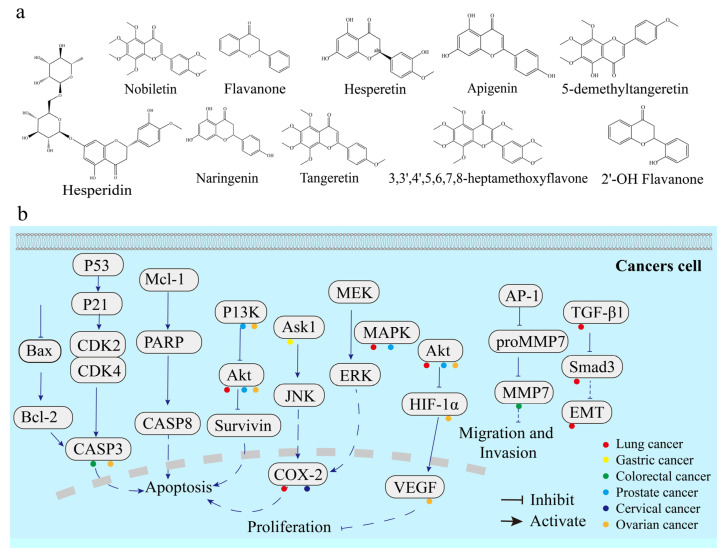
Compounds and molecular mechanisms of citrus flavonoids against non-breast cancers. (**a**) Citrus flavonoids were used to treat other cancers. (**b**) The possible molecular mechanisms of citrus flavonoids in the treatment of other cancers.

**Figure 8 antioxidants-12-00669-f008:**
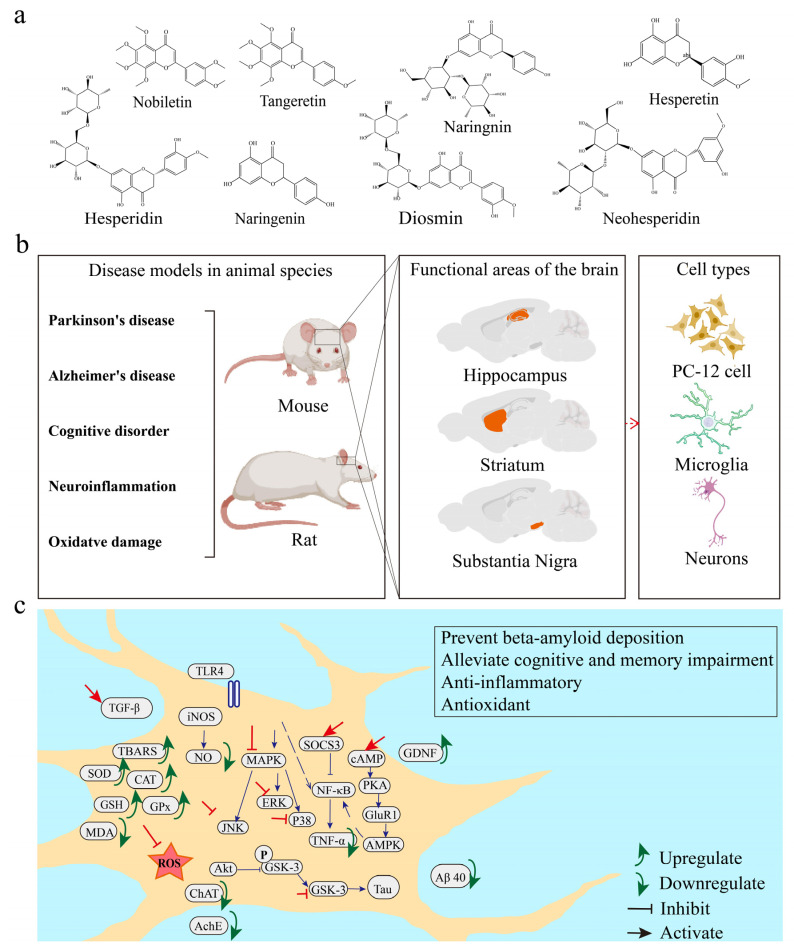
Neuroprotective effects of citrus flavonoids. (**a**), Citrus flavonoid compounds with neuroprotective effects. (**b**) Animal models, brain regions, and cell types were used to study the neuroprotective effects of citrus flavonoids. (**c**) The possible molecular mechanism of the neuroprotective effects of citrus flavonoids. TGF-β: Transforming growth factor; SOD: Superoxide dismutase; TBARS: Thiobarbituric acid-reactive substances; CAT: catalase; GSH: L-Glutathione; MDA: Malondialdehyde; GPx: Glutathione peroxidase; ROS: Reactive oxygen species; TLR4: Toll-like receptor 4; iNOS: inducible nitric oxide synthase; NO: Nitrous oxide; ERK: Extracellular regulated protein kinases; P38: mitogen-activated protein kinases p38; ChAT: Choline acetyltransferase; AchE: Acetyl cholinesterase; GSK-3: Glycogen synthase kinase-3; Tau: Tau proteins; SOCS3: Suppressor of cytokine signaling 3; NF-κB: Nuclear factor kappa-B; TNF-α: Tumor necrosis factor; cAMP: Cyclic adenosine monophosphate; PKA: Protein kinase A; GluR1: Glutamate receptor; AMPK: Adenosine 5′-monophosphate (AMP)-activated protein kinase; GDNF: glial cell-derived neurotrophic factor; Aβ 40: β-amyloid 40; Akt: Protein kinase B; JNK: c-jun n-terminal kinase; MAPK: mitogen-activated protein kinase.

**Table 1 antioxidants-12-00669-t001:** The top 25 journals related to Citrus flavonoids.

Rank	Number of Publication	% of Total Publication	Journal			
			Name	Country/Region	Impact Factor (5 Years)	H-Index (2020–2021)
1	188	5.866	*Journal of Agricultural and Food Chemistry*	United States	5.105	297.0
2	156	4.867	*Food Chemistry*	United Kingdom	7.341	262.0
3	75	2.34	*Molecules*	Switzerland	4.189	149.0
4	47	1.466	*Food & Function*	United Kingdom	6.317	89.0
5	40	1.248	*Journal of Functional Foods*	United Kingdom	4.432	84.0
6	40	1.248	*Journal of The Science of Food and Agriculture*	United Kingdom	3.547	142.0
7	38	1.186	*LWT Food Science and Technology*	United States	4.991	133.0
8	29	0.905	*Nutrients*	Switzerland	5.644	115.0
9	28	0.874	*Scientia Horticulturae*	Netherlands	3.612	112.0
10	27	0.842	*Molecular Nutrition Food Research*	Germany	6.575	141.0
11	26	0.811	*British Journal of Nutrition*	United Kingdom	3.59	188.0
12	25	0.78	*Food Research International*	United Kingdom	6.355	163.0
13	25	0.78	*Food Science and Biotechnology*	South Korea	2.468	38.0
14	25	0.78	*International Journal of Molecular Sciences*	Switzerland	5.708	162.0
15	24	0.749	*European Food Research and Technology*	Germany	2.898	102.0
16	24	0.749	*Journal of Food Processing and Preservation*	United States	2.1	48.0
17	24	0.749	*Journal of Pharmaceutical and Biomedical Analysis*	Netherlands	3.663	127.0
18	23	0.718	*Food and Chemical Toxicology*	United Kingdom	4.449	172.0
19	22	0.686	*Bioscience Biotechnology and Biochemistry*	United Kingdom	1.913	116.0
20	22	0.686	*European Journal of Pharmacology*	Netherlands	4.064	180.0
21	22	0.686	*Journal of Chromatography A*	Netherlands	4.321	229.0
22	22	0.686	*Journal of Separation Science*	Germany	3.201	102.0
23	22	0.686	*Life Sciences*	United States	4.615	164.0
24	22	0.686	*Phytotherapy Research*	United Kingdom	5.374	129.0
25	21	0.655	*Antioxidants*	Switzerland	6.084	46.0

**Table 2 antioxidants-12-00669-t002:** The top 25 co-cited journals related to citrus flavonoids.

Rank	Cited Number	Journal			
		Name	Country/Region	Impact Factor (2020–2021)	H-Index (202–2021)
1	2157	*Journal of Agricultural and Food Chemistry*	United States	5.105	297.0
2	1454	*Food Chemistry*	United Kingdom	7.341	262.0
3	703	*Molecules*	Switzerland	4.189	149.0
4	680	*Journal of Biological Chemistry*	United States	4.562	513.0
5	611	*Life Sciences*	United States	4.615	164.0
6	611	*Journal of Nutrition*	United States	4.019	265.0
7	607	*PLoS ONE*	United States	3.272	332.0
8	605	*Phytochemistry*	United Kingdom	3.814	176.0
9	582	*Journal of the Science of Food and Agriculture*	United Kingdom	3.547	142.0
10	566	*Food and Chemical Toxicology*	United Kingdom	4.449	172.0
11	553	*Biochemical Pharmacology*	United States	5.494	198.0
12	550	*Free Radical Biology and Medicine*	United States	265.0	265.0
13	548	*Phytotherapy Research*	United Kingdom	5.374	129.0
14	509	*Nature*	United Kingdom	49.962	1226.0
15	507	*Proceedings of the National Academy of Sciences of the United States of America*	United States	12.779	805
16	500	*Biochemical and Biophysical Research Communications*	United States	3.575	271
17	494	*Bioscience, Biotechnology, and Biochemistry*	United Kingdom	2.337	123
18	481	*American Journal of Clinical Nutrition*	United States	7.045	351
19	463	*Food Research International*	United Kingdom	6.475	177
20	458	*Journal of Chromatography A*	Netherlands	4.049	237
21	444	*Journal of Food Science*	United States	2.470	160
22	428	*Analytical Biochemistry*	United States	3.365	195
23	425	*Journal of Pharmaceutical and Biomedical Analysis*	Netherlands	3.571	133
24	385	*Journal of Ethnopharmacology*	Ireland	3.690	205
25	383	*Biological and Pharmaceutical Bulletin*	Japan	2.233	122

**Table 3 antioxidants-12-00669-t003:** The top 15 cited articles related to citrus flavonoids.

Rank	Times Cited	Journal				References
		Year	Name	Country	Impact Factor (2021)	
1	457	2000	*Food Chemistry*	United Kingdom	9.231	[13]
2	441	2010	*Food Chemistry*	United Kingdom	9.231	[14]
3	431	2007	*Journal of Nutrition*	United States	4.687	[15]
4	410	2006	*Bioscience, Biotechnology, and Biochemistry*	United Kingdom	2.337	[16]
5	390	2001	*Journal of Nutrition*	United States	4.687	[17]
6	382	1996	*Nutrition and Cancer*	United States	2.816	[18]
7	378	2003	*Journal of Neurochemistry*	United Kingdom	5.546	[19]
8	322	2009	*Pakistan Journal of Pharmaceutical Sciences*	Pakistan	0.863	[20]
9	306	2003	*European Journal of Clinical Nutrition*	United Kingdom	4.884	[21]
10	301	2007	*Food Chemistry*	United Kingdom	9.231	[22]
11	298	2001	*Food Chemistry*	United Kingdom	9.231	[23]
12	296	2005	*Journal of Agricultural and Food Chemistry*	United States	5.105	[24]
13	292	2006	*Food Chemistry*	United Kingdom	9.231	[25]
14	284	1999	*Journal of Agricultural and Food Chemistry*	United States	5.105	[26]
15	280	2013	*Green Chemistry*	United Kingdom	11.034	[27]

**Table 4 antioxidants-12-00669-t004:** The top 15 co-cited references related to Citrus flavonoids.

Rank	Times Cited	Journal				References
		Year	Name	Country	Impact Factor (2021)	
1	113	2008	*Journal of Agricultural and Food Chemistry*	United States	5.105	[28]
2	106	2007	*Food Chemistry*	United Kingdom	9.231	[29]
3	96	2015	*Phytotherapy Research*	United Kingdom	6.388	[30]
4	78	2006	*Bioscience, Biotechnology, and Biochemistry*	United Kingdom	2.337	[16]
5	75	2014	*Advances in Nutrition*	United States	11.567	[31]
6	68	2007	*Molecules*	Switzerland	4.189	[32]
7	59	2016	*Food Chemistry*	United Kingdom	9.231	[33]
8	58	2014	*Journal of Food Composition and Analysis*	United States	4.520	[34]
9	49	2014	*Food Chemistry*	United Kingdom	9.231	[35]
10	46	2013	*Current Opinion in Lipidology*	United States	4.616	[36]
11	43	2011	*American Journal of Clinical Nutrition*	United States	8.472	[37]
12	42	2001	*Current Medicinal Chemistry*	United Arab Emirates	4.740	[38]
13	42	2003	*European Journal of Clinical Nutrition*	United Kingdom	4.884	[21]
14	41	2001	*Journal of Nutrition*	United States	4.687	[17]
15	38	2008	*Food Chemistry*	United Kingdom	9.231	[39]

## Data Availability

Data sharing not applicable.

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
