# Peer review of "A Scientometric Study to a Critical Review on Promising Anticancer and Neuroprotective Compounds: Citrus Flavonoids"

_antioxidants, 2023, doi:10.3390/antiox12030669_

Round 1

Reviewer 1 Report

The present review aims to summarize the scientific literature on citrus-derived flavonoids highlighting the impact of that research field at bibliometric point of view. Moreover, authors described the principal known anticancer and neuro-protective effects of these compounds, together with their beneficial effects against diabetes and metabolic syndrome. Even if there are other reviews in the same field published in recent years, the scientometric details and the focus on the different specific cancer types makes this review particularly interesting for the biomedical research. However, I think that a partial revision of the text and figures is needed to make this manuscript clear and suitable for publication.

 Specific points

 1) The paragraph entitled "3.3. Citrus flavonoids and other cancers" does not contain significant information and need to be erased. The authors described the effects of citrus-derived flavonoids on the different cancer types in the paragraphs after the 3.3.

 2) Even the figure 7, mentioned in paragraph 3.3., appears to me misleading: it has the intention to sum-up all the effects of citrus-derived flavonoids on non-breast cancers. The result is a confused information on the anticancer roles of citrus-derived flavonoids.  I suggest to modify the image performing different graphical representation of the known effects of flavonoids from citrus specifying the cancer type ( colorectal, gastric, lung liver... and so on).

 3) The adaptor molecules described in the figures are in boxes of different colors: is there a meaning for the different colors? If there is a meaning it should be explained. If there is not a meaning this graphical representation results misleading.

 4) The graphical representation that summarizes the citrus-flavonoids effects on metabolic diseases is missing and needs to be added as an additional figure (figure 9).

Reviewer 2 Report

The manuscript titled “A scientometric study to a critical review on the promising anticancer and neuroprotective compounds: Citrus flavonoids” attempted to review a broad area of natural products and two unrelated disease conditions.  

 There are some deficiencies in the manuscript which need to be addressed to make it publishable:

1.      Authors might want to reconsider if they will keep two broad disease areas or focus on one.

2.      The abstract and introduction do not have the purpose of this manuscript described which is important.

3.      Check for typographic errors, example, figure 1- “donwload”

4.      Though the manuscript has few interesting highlights like journal citations and geographical distribution etc in the first part, the paper doesn’t connect well as it transitions into the mechanistic parts. Author should consider splitting this into two manuscripts- one with scientometric study and another one with mechanistic discussions.

5.      Clarity of figures 2 and 3 should be evaluated.

6.      Overall, the review lacks depth and critical analyses.

Reviewer 3 Report

This is nice work which collects most of the data concerning the anticancer and neuroprotective activity of citrus flavonoids available in the scientific literature. For me first, the part of the manuscript where the analysis of bibliometric data, is not interesting,  however, I admit it looks attractive and I understand it may be interesting for other readers. The part describing biological activity is more interesting it is concise and well written. One subject which is lacking however which should be described here is the impact of citrus flavonoids on CYP3A4 expression and activity in the context of their anticancer and neuroprotective activity. Could the Authors include a short paragraph describing this problem? 

Round 2

Reviewer 1 Report

Authors fully addressed all the points raised. In my opinion the manuscript is now acceptable for publication in the present form.

Reviewer 2 Report

Comments are reasonably addressed. Thanks